# Quantifying auditory impressions in dreams in order to assess the relevance of dreaming as a model for psychosis

**Roar Fosse** [1]*, **Frank Larøi** [2,3,4]

**1** Division of Mental Health and Addiction, Vestre Viken Hospital Trust, Drammen, Norway, **2** Department of Biological and Medical Psychology, Faculty of Psychology, University of Bergen, Bergen, Norway, **3** NORMENT – Norwegian Center of Excellence for Mental Disorders Research, University of Oslo, Oslo, Norway, **4** Psychology and Neuroscience of Cognition Research Unit, University of Liège, Liège, Belgium

* roar.fosse@vestreviken.no

## Abstract

A long noted hypothesis is that mechanisms of dreaming play a role in psychotic hallucinations. One challenge for this hypothesis is that while psychotic hallucinations primarily are auditory, dreaming most characteristically is visual. At the same time, previous studies have not explicitly examined auditory impressions in dreaming. Here, we mapped the prevalence and characteristics of auditory impressions in 130 dreams reported after spontaneous awakenings from sleep in 13 normal, healthy people. We instructed participants to report any dream they could recall and to pay particular attention to possible auditory impressions. The participants reported auditory impressions in 93.9% of their dreams on average. The most prevalent auditory type was other people speaking (83.9% of participants' dreams), followed by the dreamer speaking (60.0%), and other types of sounds (e.g. music, 33.1%). Of altogether 407 instances of auditory impressions in the 130 dreams, auditory quality was judged comparable to waking in 46.4%, indeterminate in 50.6%, and absent or only thought-like in 2.9%. The results suggest that also internally generated auditory (verbal) sensations are a central component of dreaming, typically occurring several times every night in normal, healthy people.

## Introduction

At least since the philosophers of ancient Greece, scholars have pointed out the analogy between madness (psychosis) and dreaming, portraying the psychotic person as a waking dreamer and the dream as a psychotic event [1, 2]. The core observation is that both dreaming and psychosis exhibit the combination of internally generated perceptual sensations and inadequate thoughts and reflection–including false beliefs and lack of insight into the internal origin of the percepts. The idea of dreaming as a model for madness was particularly often discussed during the nineteenth century, with scholars from various disciplines pointing to similarities in the free-flowing of thoughts, associations and combinations of objects and ideas in one's mind; strange deviations from the accustomed sober paths of associations; impressions of profound insight and enlightenment; an inability to control the often rapid train of images and ideas by an act of will; and endogenously generated percepts that were of the same type [3–7]. Brierre de Boismont [3] likened the endogenous percepts in both

**Data Availability Statement:** All relevant data are within the paper and its Supporting Information files.

**Funding:** The authors received no specific funding for this work.

**Competing interests:** The authors have declared that no competing interests exist.

dreaming and madness to a chariot drawn by horses, but without a guide, due to the horse-man being respectively asleep and insane. Laycock [4] suggested that for the insane hearing voices, there exists a state of double consciousness, in which a part of the cerebrum is in a dream, within an overall state of wakefulness. In recent days, studies employing psychometric instruments have added evidence that dreams in normally healthy people are reminiscent of waking thoughts in psychotic patients in terms of bizarre content and quasipsychotic experiences [8–10]. Researchers now also have pointed to similarities between dreaming and psychosis in neurophysiological underpinnings. This includes activation of secondary perceptual association regions in the posterior cerebral cortex, which co-occur with the internally generated percepts in the two conditions [11–14]. Moreover, dampened activity in both the dorsolateral prefrontal and posterior dorsomedial parietal cortex and reduced functional connectivity between frontal and posterior cortical regions, all may contribute to the reduced insight and inadequate reflective thought in both dreaming and psychosis [15]. In addition, similarities may be seen in the dopamine system, which is thought to be highly sensitive in psychosis [16]. In REM sleep, where the most intense dreaming takes place, the dopamine system is activated strongly and selectively among the aminergic neuronal systems [1, 17].

However, the idea of dreaming as a model for psychosis has remained speculative and controversial. One important reason for this is the apparent unequal involvement of perceptual modalities in the two conditions, with a dominance of hallucinated voices in psychosis but visual impressions in dreaming [18–20]. Among people with enduring psychosis ("schizophrenia"), 60% to 80% report a lifetime prevalence of hallucinated voices, which exceeds the 25% to 40% typically reported for visual hallucinations [21–26]. For dreaming, the evidence indicates an opposite situation, with nearly all participants reporting visual impressions in 90%-100% of their (REM sleep) dreams, typically encompassing composite visuospatial scenarios that change dynamically over time [27, 28]. In contrast, since the origin of modern dream research in the 1950s only two studies appear to have addressed auditory dream percepts. In these, no indications of endogenous auditory sensations were found in up to half of all dreams, and presumably with only limited presence in the other half [29, 30].

We asked whether auditory impressions truly are infrequent in dreams, or whether this view rather may reflect the use of imprecise methods for this domain of mental experience during sleep. In the two relevant studies that we identified in the literature, the researchers did not instruct their participants to attend to and report this type of impressions. Instead, they had participants provide general reports of dreams, with the researchers themselves then scoring these reports for the presence of various types of dream content, including auditory elements. Here, the problem of underestimation would occur if people who are not explicitly instructed to attend to and report their auditory dream impressions, fail to do so even when such impressions take place. This type of methodological bias has been demonstrated for other dream elements, such as emotions [31]. By wanting to remedy and reopen this issue, we directly and extensively examined auditory impressions in naturalistic dreams, using explicit instructions for the participants to attend to and report the presence and qualities of any such impressions.

## Methods

In this mixed qualitative–quantitative study, we combined a naturalistic approach, where participants slept at home in their own bed and reported dreams upon spontaneous awakenings from sleep, with specific instructions to attend to and report the dream quality of interest (auditory impressions).

## Participants

We used social network sampling and an advertisement on a psychology group on Facebook to recruit participants who self-reported to have good dream recall in everyday life, but with no psychiatric or severe somatic disorders, and who did not use medications that influence sleep. Eighteen participants responded and consented to participate. We excluded five participants who did not complete the study period or did not follow the dream reporting instructions. The remaining participant group included 12 females and 1 male, mean age 28.2 years (SD = 9.0).

## Variables and measures

We instructed participants to write down any dreams they could recall that took place just prior to awakening from sleep, and that they should continue until they had reported 10 dreams. For each reported dream, we asked participants to include as much detail as possible about the setting where they were, what they saw, heard, did, felt and were thinking. We stressed that they should report any dream regardless of whether it included auditory content, since the prevalence of such content was one of the research questions. At the same time, participants were asked to focus on any possible auditory content and describe this in as much detail as possible when it was present, including if it was "heard" or if it was more thought-like. When a dream contained no auditory impressions, this was to be stated.

**Types of auditory impressions.** Based on previous studies where dream content had been categorized [29, 32], we sorted auditory impressions into three types: (i) the dreamer herself speaking, (ii) speaking by other people, and (iii) other types of auditory impressions (e.g. music). We counted shouting, singing, laughing etc. as instances of speaking. We scored any dialog between the dreamer and another dream character, regardless of its length, as (only) one instance of the dreamer speaking and one instance of other people speaking.

**Qualities of auditory impressions.** We used three steps to characterize the auditory quality of the reported instances of (possible) auditory impressions. First, we inspected all auditory instances to identify similarities in the way the participants had described them. This led to the induction of five "information categories", referring to the type of information about auditory quality that the participants had conveyed: (1) "clearly heard" or described as "being like speech/sounds in a waking state", (2) thought-like or "only in thoughts" rather than auditory, or that one "just knew" something was said but without hearing it, (3) the use of adjectives for auditory qualities, such as loud, clear, or—regarding speech–self-confident, satisfied, and happy, or a clear statement that it was the voice of a named person speaking (i.e. not only "John said" but "John said, in John's voice", (4) for speech—a description only of who was speaking (e.g. "John said" or "I said something but cannot recall what"), and (5) nothing was stated about auditory qualities or the participant reported that she could not recall any qualities. In the second step, we sorted each instance of auditory impression into the information category with the lowest category number (from 1 through 5) for which it qualified. In the third and final step, we started out by considering that categories 1 and 2 conveyed unequivocal information about, respectively, "auditory quality as in waking" and "no auditory quality". Moreover, we considered category 3 as suggestive of true auditory quality. On this basis, in the result part, we treated auditory impressions sorted into either category 1 or category 3 suggestively to have a true auditory quality. In contrast, we defined as indeterminate any instances sorted into category 4 or category 5.

## Data analysis

Data presentation is mainly descriptive, with means and standard deviations and illustrations in Tables and Figures. We tested differences in the occurrence of the three types of auditory

content using paired sample t-tests, with participant as the unit of analysis. We carried out the analyses in SPSS v. 25.

### Ethics

The Regional Committee for Medical and Health Research Ethics considered the study to fall outside the area covered by the Norwegian Health research regulations and hence their mandate. The Data protection services at Vestre Viken hospital trust approved the study, 16/00117-56. Participants provided written consent.

## Results

Each of the 13 participants reported between 12 and 61 auditory instances in their 10 dreams, with a participant mean of 31.2 instances (SD = 14.8). All participants considered together, a total of 407 auditory instances were identified in 122 of the 130 dreams, with a mean of 3.1 instance per dream (SD = 2.4, median = 2) (Fig 1). Of the three auditory content types, the most frequent was other people speaking (226 instances in 109 reports), followed by the dreamer speaking (122 instances in 77 reports), and other types of sounds (59 instances in 44 reports). Of altogether 348 instances of speech (dreamer or other people speaking), what was said was cited verbatim by the dreamer in 171. In another 89 instances, the dreamer described only the general theme of the utterance, while in 66 instances s/he only reported that something was said. Another five instances were described as speech in a foreign language not understood by the dreamer, and 17 instances were not strictly speech but instead laughing, singing, screaming, cheering, or shouting. The 59 instances with sound impressions other than

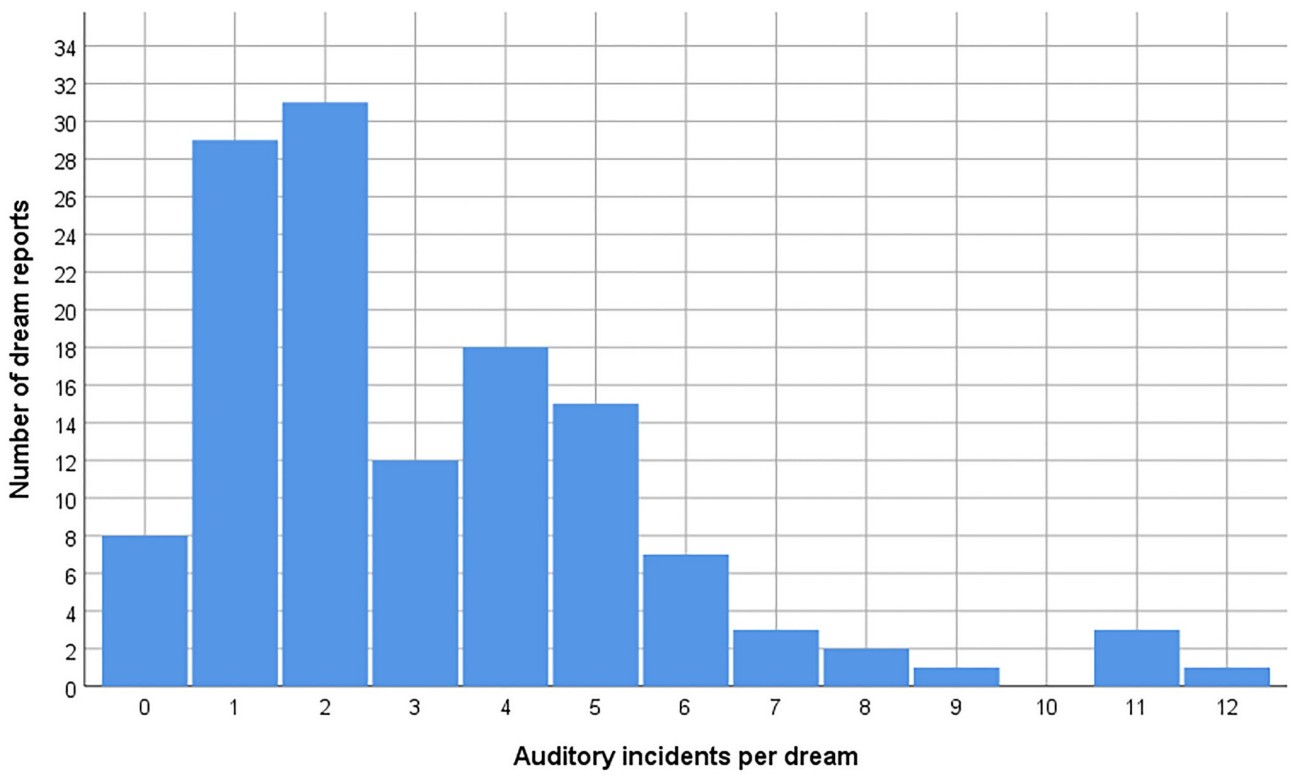

**Fig 1. Number of auditory incidents in 130 dreams.**

speech included music (e.g. piano playing, melodies from a radio, n = 15), sounds of weapons (e.g. gun shots, n = 7), vehicles (e.g. cars, motorcycles, n = 6), animals (e.g. dog barking, n = 6), objects (e.g. glass breaking, n = 7), electrical devices (e.g. a radio, an alarm, video game, n = 6), and people walking, running or skiing (n = 9), and with three instances described as "noise".

Each participant reported auditory content in 80% to 100% of their 10 dreams, mean 93.9% of the dreams (SD = 8.7) (Fig 2). Other people speaking (participant mean of 83.9%, SD = 18.1) was more prevalent than both the dreamer speaking (mean 60.0%, SD = 17.3), $t(12)$ = 3.8, p = .002, and other types of sounds (mean 33.1%, SD = 10.3), $t(12)$ = 7.3, p < .001, and the dreamer speaking was more prevalent than other types of sounds, $t(12)$ = 3.5, p = .004.

Table 1 summarizes the categorization of all 407 (possible) auditory instances with respect to information provided by the dreamer that was relevant to decide upon perceptual quality. When using our first of the five categories (clear sound/heard as in waking) and the third category (words were used to describe sound quality) to suggest the presence of true perceptual/ sound quality, 189 instances (46.4%) qualified. Slightly above half of all incidents (n = 206, 50.4%) were indeterminate in terms of perceptual quality, with descriptions being limited to who said something (fourth category) or that nothing was stated (fifth category). Only 12 instances (2.9%) were scored as lacking any sound quality, being experienced only as thoughts (category two).

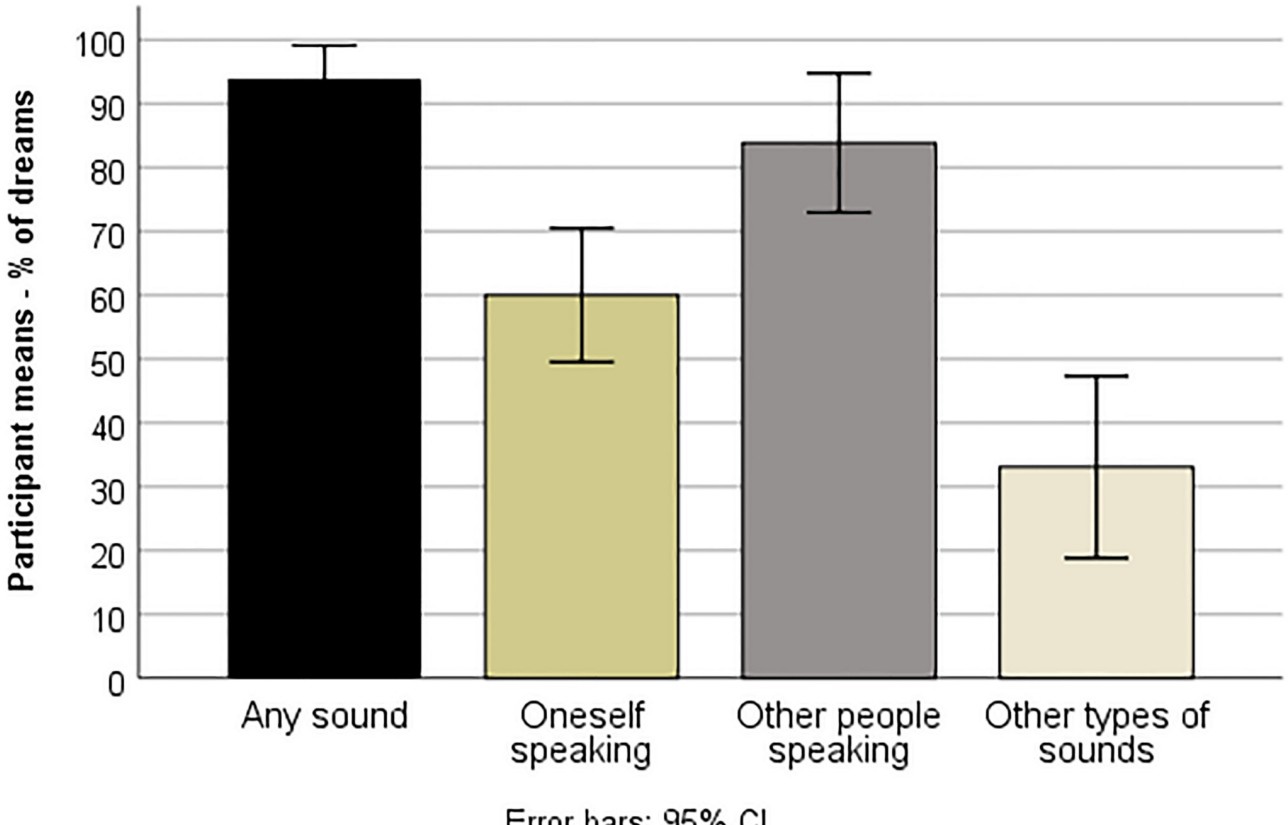

**Fig 2. Percentage of dreams with auditory content—Average across 13 participants.**

**Table 1. Auditory qualities of speech and other sounds.**

| | n | Information about auditory quality | | | | |
|---|---|---|---|---|---|---|
| | | *Was heard/clear/ as in normal life/ sound of | Only/ rather thoughts/ no sound | Description of sound/ voice quality | Described who said it (dreamer or other people) | Nothing stated/ can't recall |
| Cited verbatim | 171 | 34 | 4 | 22 | 107 | 4 |
| Only theme noted | 89 | 17 | 7 | 13 | 51 | 1 |
| No theme given[&] | 66 | 28 | 1 | 8 | 25 | 4 |
| Foreign language[+] | 5 | 2 | - | 1 | 2 | - |
| Laughing, singing etc | 17 | 10 | - | 3 | 4 | - |
| Other sounds | 59 | 37 | - | 14 | n/a | 8 |
| Total (%) | 407 | 128 (31.4%) | 12 (2.9%) | 61 (15.0%) | 189 (46.4%) | 17 (4.2%) |

*Categorization was forced and mutually exclusive, in the order from left to right for the five columns (categories 1 to 5).

[+]Speech in a foreign language not understood by the dreamer.

[&]Participants did not describe any theme, often stating that they could not recall what was said

## Discussion

A challenge for the hypothesis that dream mechanisms are relevant to psychotic hallucinations is that while psychotic hallucinations most typically are auditory, dreaming is pervasively visual. However, auditory impressions in dreams have remained largely unexplored. In the first study that examined the auditory domain in a direct and detailed manner, we found that each of 13 participants reported auditory impressions in the great majority ($\geq$ 80%) of their dreams.

To our knowledge, from 1950 and onwards, only two previous studies have reported data on (possible) auditory impressions in dreams. In each of these, external judges inspected participants' written dream reports and categorized content elements into various types. Using this method, Zadra et al. [29] conveyed that "unambiguous references to auditory impressions" were evident in 53% of 3372 transcribed home dreams from 164 participants. When analyzing 104 transcribed laboratory dreams from REM sleep in 14 participants, McCarley & Hoffman [30] identified descriptions of "auditory experiences" in 64%. In addition, three older studies from the era of introspectionism more than 100 years ago, provided data on auditory content based on intense studies of dreams in small sets of participants. Using various definitions and procedures, the authors of these studies reported that 53%, 69% and 92%, respectively, of the dreams included auditory content [33–35]. Combined with the 93% that we identified when having our participants focus explicitly on this perceptual modality, we suggest that the great majority of recalled dreams do include auditory impressions. We also found that typical for recalled dreams are multiple rather than single incidents of auditory impressions, with half of the dreams in our sample containing three or more auditory incidents, most often involving speech. Indeed, this might be as expected given prior observations that social interactions are highly characteristic of dreaming, since speech is the major form of social interaction [27, 28, 32]. From inspecting transcripts of 635 dreams obtained after forced awakenings from REM sleep in the sleep laboratory, Snyder [27] commented that dream narratives almost always involve the self together with others, usually in some manner of interaction, with descriptions of talking and verbal exchange evident in 86% to 100% of medium to long dream reports. Considering that dream reports include experiences from only the last minutes of sleep before

awakening, and that dreaming normally takes place throughout REM sleep as well as in light NREM sleep, the available evidence suggests that normal, healthy people usually experience internally generated auditory sensations an array of times every night.

While we estimated that close to half of all auditory impressions were "heard" in the dream, the acoustic qualities of the second half could not be, or were not, described, and a small subset apparently were only thought-like. This variety of auditory qualities may resemble that seen for hallucinations in psychosis, which include instances that are literally auditory, a mix of auditory and thought-like, and exclusively thought-like [36–41]. Already since Kraepelin, it repeatedly has been reported that qualities of verbal hallucinations vary and often are difficult to describe, with variants including «resonant voices», «voices of conscience», «voices which do not speak with words», «an inward voice in the thoughts, «something between hearing and foreboding», and «thoughts circulating in the air» [36, 39, 42]. A more intensive probing is required to determine the degree to which the same is the case for auditory impressions in dreams.

In comparing the phenomenology of dreaming and psychosis, both conditions appear to support auditory as well as visual endogenous percepts. Moreover, both types of internally generated sensations appear to be more frequent and typical for normal dreaming than for psychosis, although with substantial variability among people with psychosis. In dreaming, we typically act entirely within an endogenous multimodal, perceptual scenario, which contrasts to the mix of internally generated and external sensations in psychosis. The clearest difference between the two conditions with respect to the visual and auditory domains arguably is that visual impressions in (REM sleep) dreaming to a larger degree are more continuously present than they are in psychosis. At the same time, it may be premature to conclude about the degree to which visual impressions differ between the two conditions. As recently noted by Toh and colleagues [43], the apparent dominance of auditory over visual hallucinations in psychosis could reflect a bias in scientific focus. In line with this possibility, prospective monitoring studies of hallucinations that used the momentary experience sampling method rather than retrospective recall, found that visual hallucinations were as typical as auditory hallucinations in patients with a schizophrenia diagnosis. In two such studies, 62% and 26% of the participants reported the experience of visual hallucinations over a period of one week, whereas 49% and 34% reported auditory hallucinations [24, 25]. Furthermore, the combined occurrence of auditory and visual hallucinations may be considerably more common in psychosis than unimodal auditory hallucinations [24, 44].

A phenomenological feature common to psychosis and dreaming is the weakening of insight and reflective thought [15]. Studies of mental activity over the wake-sleep cycle indicate that reflective thought changes in a reciprocal, inverse manner to internally generated percepts in normal healthy people. When comparing mental activity from quiet waking (with eyes closed) to sleep onset, light NREM sleep and REM sleep, a gradual decrease is seen in reflective, directed thought, which is opposite to a gradual increase in endogenous percepts [45, 46]. A similar inverse, opposite change in these two broad classes of mental activity is also seen from early to late parts of the night within each of REM sleep and light NREM sleep [47], and from epochs without rapid eye movements ("tonic" epochs) to epochs with much rapid eye movements ("phasic" epochs) within REM sleep [48]. This characteristic of the normal wake-sleep cycle has its parallel in theories of psychotic hallucinations that focus on changes in the cognitive control over perceptual activity. Here, impairment in higher order, "top-down" and/or executive (neuro)cognitive systems (typically in the prefrontal cortex) is seen to disinhibit perceptual systems, facilitating the internal generation of sensory-like, hallucinatory experiences [14, 49, 50]. It could possibly be an intrinsic property of the brain-mind system that reduced or compromised higher order cognitive, executive control inversely facilitates multimodal

perceptual activity. Further investigations into the analogy of psychotic hallucinations and dreaming may focus on neurobiological details of changes in inhibitory neurocognitive control as well as, naturally, the complementary "bottom up" processes that contribute to facilitation of multimodal perceptual activity.

## Strengths and limitations

The spontaneous rather than instrumental awakening protocol for dream reporting that we used may have led to biases in the included set of dreams. The spontaneous awakening paradigm is generally associated with an overrepresentation of dreams with salient, dramatic, and anxious content [31, 51]. Spontaneous awakenings from sleep also often are gradual and take more time than instrumental awakenings, with the possible consequence of lost recall. It is not clear how these issues may have influenced the results. Moreover, our awakening procedure was uninformative of sleep stage. However, previous studies found that 60%-70% of spontaneously recalled dreams resulted from REM sleep [52, 53]. Subsequent studies should use dream sampling with auditory probing after instrumental awakenings from EEG-defined sleep stages. Strengths of the study include its naturalistic, ecologically valid design, and the explicit instructions to participants to report the phenomenological feature at stake.

## Conclusions

The internal generation of auditory sensations, most notably of speech, may be a typical, integrated characteristic of dreaming. The findings on auditory impressions in dreams contribute to making clear the comparative phenomenology that models of common underlying mechanisms in dreaming and psychosis must account for.

## Supporting information

**S1 Dataset.**
(XLSX)

## Acknowledgments

Thanks to Kristin Rantanen for dedicated help in recruiting participants and carrying out the study.

## Author Contributions

**Conceptualization:** Roar Fosse, Frank Larøi.

**Data curation:** Roar Fosse.

**Formal analysis:** Roar Fosse.

**Investigation:** Roar Fosse.

**Methodology:** Roar Fosse, Frank Larøi.

**Project administration:** Roar Fosse.

**Writing – original draft:** Roar Fosse.

**Writing – review & editing:** Roar Fosse, Frank Larøi.

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
