## [Decision Letter · Decision Letter 0]

13 Jan 2020

PONE-D-19-29802

Reconsidering the relevance of dreaming for psychotic hallucinations: Quantifying auditory impressions in dreams

PLOS ONE

Dear Dr. Fosse,

Thank you for submitting your manuscript to PLOS ONE. After careful consideration, we feel that it has merit but does not fully meet PLOS ONE’s publication criteria as it currently stands. Therefore, we invite you to submit a revised version of the manuscript that addresses the points raised during the review process.

Please address all reviewer comments. In addition to addressing Reviewer 1's comments, this will include revising the introduction/title/results to focus the paper more on what was done (assessment of auditory component), and saving comparisons to psychosis to the discussion section.

We would appreciate receiving your revised manuscript by Feb 27 2020 11:59PM. To enhance the reproducibility of your results, we recommend that if applicable you deposit your laboratory protocols in protocols.io, where a protocol can be assigned its own identifier (DOI) such that it can be cited independently in the future. For instructions see: http://journals.plos.org/plosone/s/submission-guidelines#loc-laboratory-protocols

We look forward to receiving your revised manuscript.

Kind regards,

Dhakshin Ramanathan, MD.,PhD,

Academic Editor

PLOS ONE

Journal Requirements:

Reviewers' comments:

Reviewer's Responses to Questions

**Comments to the Author**

1. Is the manuscript technically sound, and do the data support the conclusions?

Reviewer #1: Yes

Reviewer #2: Partly

2. Has the statistical analysis been performed appropriately and rigorously? 

Reviewer #1: Yes

Reviewer #2: N/A

3. Have the authors made all data underlying the findings in their manuscript fully available?

Reviewer #1: Yes

Reviewer #2: Yes

4. Is the manuscript presented in an intelligible fashion and written in standard English?

Reviewer #1: Yes

Reviewer #2: Yes

5. Review Comments to the Author

Reviewer #1: Lines 66 & 69 seem to be repeated in lines 71-72. Could these be merged?

Line 107, three categories of auditory content are said here to be subject to matched t-tests. But line 97 referred also to thought-like auditory content. Should that not also be a category for these t-tests?

The auditory content categories are 1 to 5, with clear auditory content being category 1 (lines 112-120). But these do not look to be ordinal categories to me. The main issue seems to be comparing 2 and 3; is it that 3 has no speech content, only form? But in some ways category 2 seems to be less auditory than category 3. Some more detail is needed to justify that this is an ordinal scale. Note that lines 119-120 do indicate that this is seen as an ordinal scale: 'We sorted each instance of auditory content into the category with the lowest category number (from 1 through 5) that it qualified for.'

Move these words to the end of the sentence on line 186:

of at least a moderate extent

Reviewer #2: The authors aim to provide a qualitative assessment of auditory content in dreams. Participants were individuals with self-reports of “good dream recall”.

Major comments:

1. Reading the introduction, it is not altogether clear what the purpose of the study is. Would suggest the authors being more direct as to the purpose of the study, which is to provide a qualitative assessment of auditory content in dreams, rather than the speculative links between dreaming and psychosis. In line with this, the title is overly assuming and should reflect the content of the study being reported, again not gross speculations.

2. The choice to compare dreams to ‘psychosis’ seems somewhat arbitrary. It is conceivable that they share some similarity in terms of neural circuitries, but the authors point out rather well that dreams and psychotic experiences are different.

Minor comments:

3. It is unclear what “specific probes” refers to in line 80.

4. It would be interesting to report visual with auditory content.

6. PLOS authors have the option to publish the peer review history of their article (what does this mean?). If published, this will include your full peer review and any attached files.

Reviewer #1: No

Reviewer #2: No

---

## [Author Response · Author response to Decision Letter 0]

20 Feb 2020

Response letter

Dear editor,

Thank you for a favorable review, with the decision that the manuscript needed minor revisions. Below, we have particularly addressed Reviewer 1’s comments, as you instructed. In addition, given your comments and those of Reviewer 2, we found it necessary to modify the Introduction so that it: (i) more clearly describes the background and rationale of the study, which is the idea that dreaming may be a model for psychosis, which, however speculative, is the sole motivation behind our study, and (ii) what we did in this study, i.e. assessed auditory experiences in dreams.

We have uploaded both a revised version of the manuscript with track changes turned on, and a clean version of the revised manuscript. Page numbers and line numbers referred to in the author reply sections below, refer to the uploaded manuscript version with track changes turned on.

Sincerely yours,

Roar Fosse

Reply to Editor’s comments

E.1. In addition to addressing Reviewer 1's comments, this will include revising the introduction/title/results to focus the paper more on what was done (assessment of auditory component), and saving comparisons to psychosis to the discussion section.

Author response: First, we have changed the title to more clearly convey the focus of the paper, see E.1, p. 1, line 1. 

Second, we have revised the last paragraph in the Introduction to clarify what we did in the study, see E.1. p. 5 lines 96 and out the paragraph. 

Third, we have revised the first paragraph in the Introduction in order to make more clear the basis for the idea/ hypothesis of dreaming as a model for psychosis. The basis first and foremost is a long history of observations that dreaming and psychosis share phenomenological similarities, and, second, more recent observations of parallels in neurophysiological underpinnings. We have now been cautious to present these similarities only in order to describe the basis for the idea of dreaming as a model for psychosis, which is the sole motivation and starting point of our study – and which also the discussion part elaborates upon. We found it pertinent to be particularly clear about the basis for the hypothesis of dreaming as a model for psychosis, given comments from Reviewer 2 that could indicate the need for a strengthened description of this basis. See the first paragraph in the Introduction, E.1 page 3, line 43 and out the paragraph. 

Fourth, we have made sure that the results section focuses entirely on auditory impressions in dreams.

 

Reply to Reviewers’ comments

Reviewer #1: 

R1.1 Lines 66 & 69 seem to be repeated in lines 71-72. Could these be merged?

Author response. We have deleted/ merged the repetitions. We also have rewritten the last part of the relevant paragraph as well as the next paragraph, based on other Reviewer comments, See R1.1 p. 4 line 90.

R1. 2. Line 107, three categories of auditory content are said here to be subject to matched t-tests. But line 97 referred also to thought-like auditory content. Should that not also be a category for these t-tests?

Author response: We only sorted auditory content into three types; dreamer speaking, other characters speaking, and other types of sound impressions than speaking. In a next step, we scored each instance of all three of these auditory types for auditory quality, referring to its sound characteristic. Here, “no sound/ only thought-like” is one of several information categories that we extracted to determine auditory quality of each instance and it is not an auditory type as such. But obviously – we were not sufficiently clear in our description of these details. We have amended the description in Methods by: (i) moving the part about auditory qualities from the data analysis section to the section for variables and measures, (ii) adding subheadings for both auditory qualities and auditory types, and (iii) specifying that auditory qualities were scored for all instances of all three types of auditory content, and we have also revised the description of the procedure we used to identify/ estimate auditory qualities. See R1.2 on page 6 line 133 and out the paragraph and R1.2 on page 7 line 140 and out the paragraph.

R1. 3. The auditory content categories are 1 to 5, with clear auditory content being category 1 (lines 112-120). But these do not look to be ordinal categories to me. The main issue seems to be comparing 2 and 3; is it that 3 has no speech content, only form? But in some ways category 2 seems to be less auditory than category 3. Some more detail is needed to justify that this is an ordinal scale. Note that lines 119-120 do indicate that this is seen as an ordinal scale: 'We sorted each instance of auditory content into the category with the lowest category number (from 1 through 5) that it qualified for.'

Author response: We apologize for having written up this section, as well as the result part about the quality variable, in a self-contradictory and confusing manner. We have revised the section in Methods so that it more precisely and coherently conveys how we went about to estimate/ identify auditory quality (and not the content) of all instances of (possible) auditory impressions that the participants reported. We now specify that (i) the five categories are really “information categories”, pertaining to the type of information that the participants provided about the auditory quality of their dream impressions (be it speech or other type of sounds), (ii) that inducing these five categories was the first step in a three-step process of deciding upon a way to present evidence about auditory quality (or estimated quality, more precisely), and (iii) we now describe a second step where we sorted all reported auditory events into one of the five information categories, and (iv) that the third and last step in our procedure was to use this categorization as a basis to estimate auditory quality for each instance as either “truly/ suggestively present”, “indeterminate”, or “absent/ no auditory quality”. In the Result section for this variable, we have rewritten the text and also changed information in the table, so that the issue now hopefully comes through in a more coherent manner. See R1.3 on page 7 lines 143 to 158 and R1.3 on page 9 from line 208 to 220 (including the table).

R1. 4. Move these words to the end of the sentence on line 186: “of at least a moderate extent”

Author response. We understand that this sentence was unclear. Unfortunately, the suggested change would alter the intended meaning of the sentence. We wanted to say that in dreams that are at least of moderate length/ duration, auditory impressions are typical. We have simplified the relevant sentence, which now reads: “Combined with the 93% that we identified when having our participants focus explicitly on this perceptual modality, we suggest that the great majority of recalled dreams do include auditory impressions.” See R1.4 on page 11, lines 241-242.

Reviewer #2: 

Major comments:

R2. 1a. Reading the introduction, it is not altogether clear what the purpose of the study is. Would suggest the authors being more direct as to the purpose of the study, which is to provide a qualitative assessment of auditory content in dreams, rather than the speculative links between dreaming and psychosis. 

Author response. We have revised the Introduction so that it (a) in the first paragraph provides a stronger presentation of the basis for the hypothesis of dreaming as a possible model for psychosis, and (b) uses the last paragraph to describe more specifically the focus and aim of the study. For the latter, see R 2.1 on page 5, lines 96-110.

R2.1b In line with this, the title is overly assuming and should reflect the content of the study being reported, again not gross speculations.

Author response. We have changed the title to better reflect the focus of the study. At the same time, given that the rationale and academic context for the study is the hypothesis of dreaming as a model for madness, we have kept a reference to this in the new title. The revised title is: “Quantifying auditory impressions in dreams in order to assess the relevance of dreaming as a model for psychosis”, see R2.1 on page 1, line 1.

R2. 2. The choice to compare dreams to ‘psychosis’ seems somewhat arbitrary. It is conceivable that they share some similarity in terms of neural circuitries, but the authors point out rather well that dreams and psychotic experiences are different.

Author response: We have revised the first paragraph of the Introduction to describe better the hypothesis of dreaming as a model for psychosis and the rationale behind this hypothesis. It is this hypothesis that is the entire motivation and context for our study. It is possible to use several pages to present the long history of this idea, however, we consider that one introductory paragraph is sufficient in this context to show how this hypothesis is far from arbitrary. See page 3 lines 43-67.

Minor comments:

R2. 3. It is unclear what “specific probes” refers to in line 80.

Author response: We have removed the word “probe”, so that it now reads “specific instructions”. See page 6, line 114.

---

## [Editor Report · Decision Letter 1]

25 Feb 2020

Quantifying auditory impressions in dreams in order to assess the relevance of dreaming as a model for psychosis

PONE-D-19-29802R1

Dear Dr. Fosse,

We are pleased to inform you that your manuscript has been judged scientifically suitable for publication and will be formally accepted for publication once it complies with all outstanding technical requirements.

With kind regards,

Dhakshin Ramanathan, MD.,PhD,

Academic Editor

PLOS ONE
---

## [Editor Report · Acceptance letter]

27 Feb 2020

PONE-D-19-29802R1 

Quantifying auditory impressions in dreams in order to assess the relevance of dreaming as a model for psychosis 

Dear Dr. Fosse:

I am pleased to inform you that your manuscript has been deemed suitable for publication in PLOS ONE. Congratulations! Your manuscript is now with our production department. 

With kind regards,

on behalf of

Dr. Dhakshin Ramanathan 

Academic Editor

PLOS ONE